

# Complexity of human walking: the attractor complexity index is sensitive to gait synchronization with visual and auditory cues

Philippe Terrier

Haute Ecole Arc Santé, HES-SO University of Applied Sciences and Arts Western Switzerland, Neuchâtel, Switzerland
Clinique romande de réadaptation SUVA, Sion, Switzerland
Department of Thoracic and Endocrine Surgery, University Hospitals of Geneva, Geneva, Switzerland

## ABSTRACT

**Background**. During steady walking, gait parameters fluctuate from one stride to another with complex fractal patterns and long-range statistical persistence. When a metronome is used to pace the gait (sensorimotor synchronization), long-range persistence is replaced by stochastic oscillations (anti-persistence). Fractal patterns present in gait fluctuations are most often analyzed using detrended fluctuation analysis (DFA). This method requires the use of a discrete times series, such as intervals between consecutive heel strikes, as an input. Recently, a new nonlinear method, the attractor complexity index (ACI), has been shown to respond to complexity changes like DFA, while being computed from continuous signals without preliminary discretization. Its use would facilitate complexity analysis from a larger variety of gait measures, such as body accelerations. The aim of this study was to further compare DFA and ACI in a treadmill experiment that induced complexity changes through sensorimotor synchronization.

**Methods**. Thirty-six healthy adults walked 30 min on an instrumented treadmill under three conditions: no cueing, auditory cueing (metronome walking), and visual cueing (stepping stones). The center-of-pressure trajectory was discretized into time series of gait parameters, after which a complexity index (scaling exponent alpha) was computed via DFA. Continuous pressure position signals were used to compute the ACI. Correlations between ACI and DFA were then analyzed. The predictive ability of DFA and ACI to differentiate between cueing and no-cueing conditions was assessed using regularized logistic regressions and areas under the receiver operating characteristic curves (AUC).

**Results**. DFA and ACI were both significantly different among the cueing conditions. DFA and ACI were correlated (Pearson's $r = 0.86$). Logistic regressions showed that DFA and ACI could differentiate between cueing/no cueing conditions with a high degree of confidence (AUC = 1.00 and 0.97, respectively).

**Conclusion**. Both DFA and ACI responded similarly to changes in cueing conditions and had comparable predictive power. This support the assumption that ACI could be used instead of DFA to assess the long-range complexity of continuous gait signals. However, future studies are needed to investigate the theoretical relationship between DFA and ACI.

Corresponding author
Philippe Terrier,
Philippe.Terrier@he-arc.ch,
ph.terrier@gmail.com

## INTRODUCTION

Gait is a stereotyped sequence of movements that enable human beings to move through their environment. A fluid and stable gait requires the complex coordination of dozens of muscles controlling multiple joints. Biomechanical and energy constraints limit the range of gait movements to a narrow window (*Holt et al., 1995*); for example, at a preferred walking speed, step length and step time vary by only a few percent (*Terrier, Turner & Schutz, 2005*). It was previously thought that these small variations were random noise introduced by residual neuromuscular inaccuracies; however, after studying the structure of gait variability among hundreds of consecutive strides, it was observed that stride-to-stride fluctuations were not totally random but instead exhibited a fractal pattern (*Hausdorff et al., 1995*). Fractal fluctuations in time series produced by living beings have been deemed to be a signature of their complex internal organization and of the feedback loops needed to adapt behaviors to environmental changes (*Goldberger et al., 2002*; *West, 2013*). Accordingly, physiological time series most often exhibit scaling properties and statistical persistence. Regarding human walking, the complex fluctuations in stride intervals, stride speeds, and stride lengths exhibit fractal patterns with inverse power-law memory (*Hausdorff et al., 1995*; *Terrier, Turner & Schutz, 2005*); that is, a change occurring at a given gait cycle can potentially influence another cycle dozens of steps later.

The fractal pattern of gait fluctuations can be disrupted by sensorimotor synchronization. It is possible for humans to synchronize their stepping with external rhythmic cues, such as walking in time with a musical rhythm (auditory cueing). In such cases, stride-to-stride fluctuations become anti-persistent; that is, stride intervals tend to oscillate stochastically around the imposed pace (*Terrier, Turner & Schutz, 2005*; *Delignières & Torre, 2009*; *Sejdić et al., 2012*; *Choi et al., 2017*). In other words, a long stride interval has a higher probability of being followed by a short stride interval. Similarly, time series of stride speeds are anti-persistent in treadmill walking, in which a constant speed is imposed by the treadmill belt (*Dingwell & Cusumano, 2010*). The fractal pattern of stride speeds can be restored using self-paced treadmills, in which the belt speed is dynamically controlled by the walking subjects (*Choi et al., 2017*). In treadmill experiments, if an additional instruction of gait synchronization is superimposed on the task of walking at the belt speed, a generalized anti-persistent pattern is then observed (*Terrier & Dériaz, 2012*; *Roerdink et al., 2015*; *Choi et al., 2017*). This phenomenon exists both when synchronizing stride intervals to a metronome (auditory cueing), and when aligning step lengths to visual cues projected onto the treadmill belt (visual cueing) (*Terrier, 2016*).

In 2010, Dingwell and Cusumano hypothesized that the emergence of anti-persistence was linked to the degree of voluntary control dedicated to the gait. They suggested that, during a normal gait, deviations go uncorrected and can persist across consecutive

strides (under-correction). In contrast, in paced walking, deviations are followed by rapid corrections that lead to anti-persistence (over-correction). This "tight control" hypothesis has been supported by other studies (*Roerdink et al., 2015*; *Bohnsack-McLagan, Cusumano & Dingwell, 2016*). Earlier this year, Roerdink et al. further demonstrated that the degree of anti-persistence can be modulated by constraining the maneuverability range on a treadmill (*Roerdink et al., 2019*). In short, characterizing the noise structure of gait variability helps us to better understand gait control; among other things, it can provide information about whether a gait is highly controlled or more automated. In addition, cued walking has important applications for rehabilitation in gait disorders (*Yoo & Kim, 2016*; *Pereira et al., 2019*).

Detrended fluctuation analysis (DFA) is typically the preferred method to identify the fluctuation structure in a time series of gait parameters. Introduced in 1995 by Hausdorff et al., DFA identifies the modification of a signal's variance at different time scales. DFA can unmask underlying fluctuation structures that may be otherwise obscured by time series trends (*Peng et al., 1995*). The presence of power-law scaling is determined through the scaling exponent alpha ($\alpha$); if the exponent is small ($\alpha < 0.5$), the fluctuations are deemed to be anti-persistent. Statistical persistence corresponds to $\alpha$ values higher than 0.5 and an $\alpha$ value equal to 0.5 indicates a random, uncorrelated noise (see Appendix B in *Terrier & Dériaz (2013)* for further information).

DFA requires a non-periodical, discrete time series as an input. Foot switches, i.e., pressure sensitive insoles, can be used to detect heel strikes on the ground and can thus collect time series of stride intervals (*Hausdorff, Ladin & Wei, 1995*; *Sejdić et al., 2012*; *Almurad et al., 2018*). Several methods using the continuous measure of the positions of various body parts have also been proposed: (1) high-accuracy GPS (*Terrier, Turner & Schutz, 2005*); (2) 3-D video analysis of treadmill walking (*Dingwell & Cusumano, 2010*); and (3) an instrumented treadmill that records the center-of-pressure trajectory (*Terrier & Dériaz, 2012*; *Terrier, 2016*; *Roerdink et al., 2019*). These methods require a preliminary discretization of the position signals via minima/maxima detection algorithms (*Terrier & Schutz, 2005*; *Roerdink et al., 2008*; *Dingwell & Cusumano, 2010*).

Other studies attempted to retrieve stride intervals from acceleration signals (*Terrier & Dériaz, 2011*), but the correct discrimination of strides can be challenging. Accelerometers are most often attached to the lower back for optimally assessing whole-body movements and for enhancing the compliance in wearing a sensor over long periods of time. The dampening of accelerations throughout the limbs can make difficult the detection of foot contacts, which are required to compute stride durations (*Terrier & Reynard, 2018*). For example, a poor detection of gait events leads to large errors when evaluating walking distance from trunk accelerations (*Lopez et al., 2008*). Although solutions exist under optimal conditions (*González et al., 2010*), it has been suggested that methods that do not require a preliminary detection of gait events could be advantageous when studying pathological gaits with atypical acceleration signals (*Riva et al., 2013*). A method that can analyze gait complexity from continuous signals may be useful in ecological monitoring of pathological gaits (*Terrier et al., 2017*).

The discrete gait time series that are analyzed through DFA are fundamentally the output of a continuous process. Indeed, gait control coordinates muscles and joints continuously during successive gait cycles; this process generates stride intervals, stride lengths, and stride speeds as outputs. It is questionable whether it is even possible to retrieve the fractal signature of long-range stride fluctuations in a continuous signal that could capture both intra- and inter- stride gait dynamics. In 2013, I hypothesized that an attractor that reflects short-term gait dynamics could also contain information about long-term gait complexity (*Terrier & Dériaz, 2013*). In 2018, I explored this hypothesis further (*Terrier & Reynard, 2018*): I proposed the use of a new gait complexity index computed from continuous signals, which I named the attractor complexity index (ACI).

ACI is a new term for long-term local dynamic stability (LDS)—also referred to as divergence exponent or lambda ($\lambda$)—which was introduced by *Dingwell et al. (2000)* and *Dingwell & Cusumano, (2000)*. This algorithm, based on Lyapunov exponents used in chaos theory (*Dingwell, 2006*; *Mochizuki & Aliberti, 2017*), has been recommended to assess gait stability and fall risk (*Bruijn et al., 2013*). LDS requires the construction of an attractor in the phase space by means of time delay embedding of continuous signals, such as body accelerations (*Takens, 1981*; *Rosenstein, Collins & De Luca, 1993*; *Terrier & Dériaz, 2013*).

LDS is defined as the divergence rate among attractor trajectories. The divergence rate can be evaluated at different intervals, either immediately after the initial separation between adjacent trajectories (short-term LDS) or several strides later (long-term LDS). In the years following Dingwell's seminal articles, it became clear that long-term LDS was in fact not a good index for predicting fall risk and gait stability (*Bruijn et al., 2013*), but that short-term LDS had better properties for gait stability analysis, as shown in modeling studies (*Su & Dingwell, 2007*; *Bruijn et al., 2012*).

Studies have shown that long-term LDS responded to various experimental conditions independently of short-term LDS. In visually and mechanically destabilizing environments, short-term and long-term LDS vary in opposite directions (*McAndrew, Wilken & Dingwell, 2011*). Similar results have been obtained when galvanic vestibular stimulation is used to impair dynamic balance (*Van Schooten et al., 2011*). Walking on a compliant surface decreases long-term LDS, with no relevant effects on short-term LDS (*Chang et al., 2010*). Overall, an accumulation of evidence supports the fact that long- and short-term LDS measure different aspects of gait control, which may justify a change in the terminology to clearly differentiate between them.

Given that long-term LDS is not a gait stability index, renaming it as ACI seems appropriate. Indeed, as demonstrated through a modelling approach, ACI is highly sensitive to the noise structure of stride intervals (*Terrier & Reynard, 2018*). More precisely, a low ACI is associated with statistical anti-persistence, and a high ACI is associated with persistence. Furthermore, it has been shown that when stride intervals are kept constant, divergence curves become flat after only two strides (see Fig. 2 in *Terrier & Reynard (2018)*). Although additional theoretical work is required to explore the causes of this sensitivity, it can be assumed that the complex gait dynamic is reflected by wider boundaries in the attractor, which allows further long-term divergence. In contrast, statistical anti-persistence signals a less complex gait dynamic, a more restricted attractor, and therefore a lower long-term

divergence rate. The fact that no divergence is observed if stride intervals are kept constant further supports this hypothesis.

The objective of the present study was to confirm that ACI can be used to assess gait complexity from continuous signals without preliminary discretization. In my 2018 study (*Terrier & Reynard, 2018*), I hybridized acceleration signals with artificial signals to explore this assumption. Here, in order to apply ACI to real signals, I computed both ACI and scaling exponents ($\alpha$s) from a center-of-pressure trajectory recorded in a treadmill experiment that submitted participants to either visual or auditory cueing. I then explored the responsiveness of ACI to the cueing conditions, as well as correlations between ACI and $\alpha$s. The ability of ACI and $\alpha$s to predict cueing conditions was also assessed. The study also had two secondary objectives: to test the appropriateness of different intervals for computing ACI, and to evaluate short-term LDS's sensitivity to cueing.

## MATERIALS & METHODS

### Data

Data from a previous study were re-analyzed (*Terrier, 2016*). In summary, 36 individuals walked for 30 min on an instrumented treadmill at their preferred speed. They were exposed to three different conditions of 10 min duration each: (1) normal walking with no cueing; (2) walking while synchronizing their gait cadence to an isochronous metronome (auditory cueing); and (3) walking while targeting visually projected shapes with their feet (visual cueing).

### Ethics statement

The present study is a re-analysis of an anonymized database and is not considered as a human research needing authorization from an ethic committee. Consent was obtained for anonymization and reuse. Please refer to the ethic statement in the original publication for further information (*Terrier, 2016*).

### Data availability

Individual data are available in the Supplemental Information. Raw signals are hosted on Figshare.

### Data processing

For each condition, 1,000 steps (500 gait cycles) were recorded. The force platform embedded into the treadmill recorded the position (Cartesian coordinates, anteroposterior [AP] and mediolateral [ML] axes) of the center of pressure at a sampling rate of 500 Hz. Based on the detection of heel strikes in the anteroposterior (AP) signal, time series of stride time (ST), stride length (SL) and stride speed (SS) were computed (*Roerdink et al., 2008*). Next, the noise structure of stride-to-stride fluctuations were assessed with DFA (for in-depth descriptions of the DFA algorithm, see *Terrier, Turner & Schutz (2005)*, *Terrier & Dériaz (2012)* and *Terrier & Dériaz (2013)*; DFA results—the scaling exponents $\alpha$—are shown in *Terrier (2016)*). DFA was implemented with box sizes ranging from 12 to 125 (i.e., N / 4) using the evenly spaced algorithm (*Almurad & Delignières, 2016*).

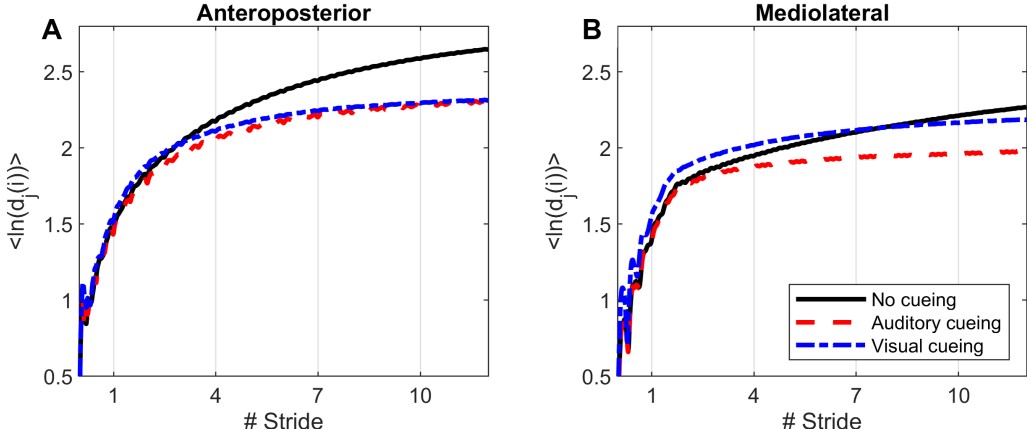

**Figure 1** **Divergence curves.** Using time-delay embedding, 5-dimensional attractors were reconstructed from the anteroposterior and mediolateral coordinates of a center-of-pressure trajectory. The logarithmic divergence from neighbor trajectories ($y$-axis) was averaged across trajectories and participants ($N = 36$), and drawn against normalized time (strides, $x$-axis). Three curves are shown, one for each experimental condition.

The 500 Hz signal from the AP and ML signals were then low-pass filtered (18 Hz 12th order Butterworth) and down-sampled to 100 Hz. Raw 500-strides signals were resampled at a constant number of 50,000 samples, i.e., 100 points per stride.

Computations of nonlinear indexes of gait stability (LDS) and complexity (ACI) were implemented via the same methods as in previous studies that used Rosenstein's algorithm (*Rosenstein, Collins & De Luca, 1993*; *Terrier & Dériaz, 2013*; *Terrier & Reynard, 2015*). High dimensional attractors were built according to the delay-embedding theorem. The average mutual information of each signal was used to assess the time delay (*Fraser & Swinney, 1986*). A common dimension of five was determined with a global false nearest neighbor analysis (*Kennel, Brown & Abarbanel, 1992*). Average divergence of the attractor was defined as $avg(ln[d_j(i)])$, that is, the logarithm of the $i$th Euclidian distance $d$ downstream of the $j$th pair of nearest neighbors in the attractor, averaged over all pairs. Time was normalized by ST. Resulting divergence curves are shown in Fig. 1. The exponential divergence rate, calculated as $avg(ln[dj(i)]) / stride$, was evaluated with linear fits across several spans as follows: 0–0.5 stride (LDS), 1–4 strides (ACI [1–4]), 4–7 strides (ACI [4–7]), and 7–10 strides (ACI [7–10]). These spans not only cover the classical range usually used for computing the long-range LDS (4–10), but also cover spans closer to initial separation that have not been studied so far (1–4).

## Statistics

Notched boxplots were used to depict the distribution of the individual results (Figs. 2 and 3). Descriptive statistics (means and standard deviations [SD]) were computed for the ACIs (Table 1). LDS statistics can be found in the Supplemental Information. Figure 4 shows the effect sizes (Hedges' $g$) of the differences between conditions (i.e., auditory cueing minus no cueing, and visual cueing minus no cueing), as well as Bonferroni corrected 95% confidence intervals.

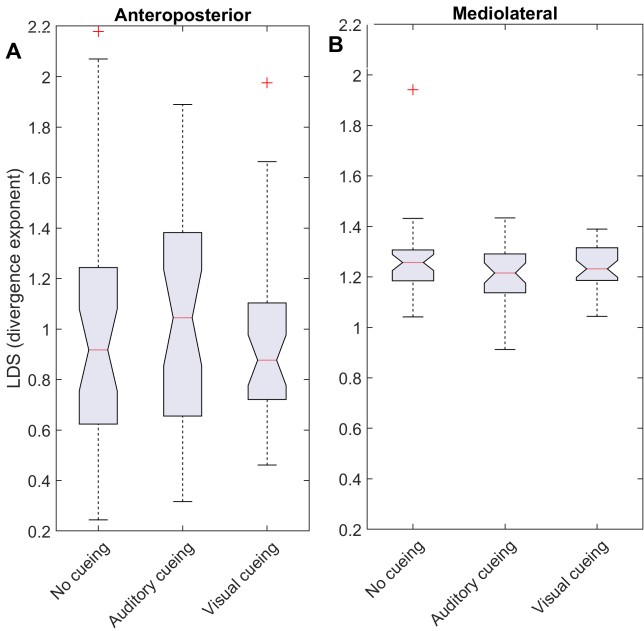

**Figure 2 Descriptive statistics of the local dynamic stability (LDS).** The notched boxplots summarize the distribution of individual results ($N = 36$) across the three experimental conditions for the anteroposterior (A) and the mediolateral (B) signals. The notch extremes correspond to the 95% confidence intervals of the medians. The red + symbols indicate outliers.

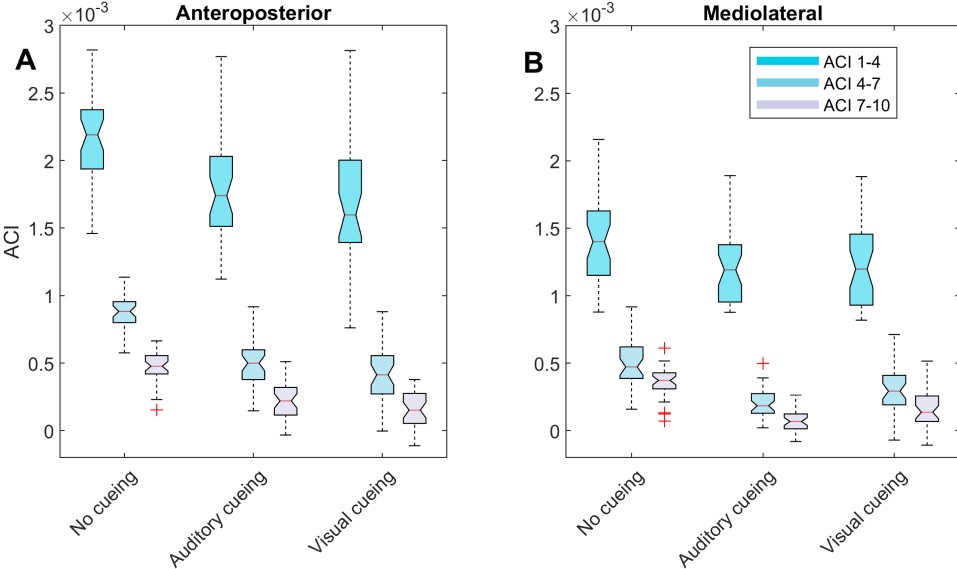

**Figure 3 Descriptive statistics of the attractor complexity index (ACI).** The notched boxplots summarize the distribution of individual results ($N = 36$) across the three experimental conditions for the three different ACI spans, and for the anteroposterior (A) and the mediolateral (B) signals. The notch extremes correspond to the 95% confidence intervals of the medians. The red + symbols indicate outliers.

**Table 1 Descriptive statistics of the attractor complexity index (ACI).** Means and standard deviations (SD) of ACI measured in the 36 subjects under the three experimental conditions. AP, anteroposterior; ML, mediolateral.

| N = 36 ACI × 1,000 | ACI [1–4] | | | | ACI [4–7] | | | | ACI [7–10] | | | |
|---|---|---|---|---|---|---|---|---|---|---|---|---|
| | AP | | ML | | AP | | ML | | AP | | ML | |
| | Mean | SD | Mean | SD | Mean | SD | Mean | SD | Mean | SD | Mean | SD |
| No cueing | 2.16 | (0.34) | 1.39 | (0.30) | 0.87 | (0.12) | 0.51 | (0.19) | 0.47 | (0.12) | 0.36 | (0.11) |
| Auditory cueing | 1.78 | (0.37) | 1.21 | (0.26) | 0.50 | (0.19) | 0.20 | (0.11) | 0.21 | (0.13) | 0.08 | (0.09) |
| Visual cueing | 1.68 | (0.52) | 1.20 | (0.29) | 0.43 | (0.22) | 0.32 | (0.19) | 0.16 | (0.13) | 0.16 | (0.14) |

The correlations among the variables are illustrated in Fig. 5 through scatter plots and linear fits. Pearson's correlation coefficients ($r$) and null hypotheses for a null correlation coefficient were also assessed.

Least absolute shrinkage and selection operator LASSO (*Tibshirani, 1996*) and logistic regressions were used to assess the extents to which DFA, LDS and ACI could differentiate between the cueing (auditory and visual) and no-cueing conditions. The LASSO algorithm had the advantage of regularizing the fit for lower overfitting and of selecting the most important predictors. The dependent binary variable was coded as no-cueing = 1 (36 observations), and cueing (auditory and visual) = 0 (72 observations). Three models were fitted as follows: Model 1: the independent variables were LDS-AP and LDS-ML (two predictors); Model 2: the independent variables were ACI [1–4], ACI [4–7], and ACI [7–10] for both the ML and AP directions (six predictors); and Model 3: the independent variables were $\alpha$-ST, $\alpha$-SL, and $\alpha$-SS (three predictors). All $\alpha$ values were taken from *Terrier (2016)*. The LASSO regularization factor was set via 10-fold cross-validation. Receiver operating characteristic (ROC) curves were used to illustrate the models' diagnostic abilities. Areas under the curves (AUCs), along with bootstrapped confidence intervals, were computed as well (Fig. 6). Sensitivity and specificity were also evaluated considering that the predicted class was 1 if the predicted probability was higher than 0.5. Figure 7 presents the standardized coefficients of the multivariable logistic models, which indicate the relative importance of each predictor.

## RESULTS

Divergence curves (Fig. 1) revealed a clear difference between cueing and no-cueing conditions, especially for the AP signal. In the no-cueing condition (black curve), divergence increased steadily, with moderate dampening. In contrast, for both auditory and visual cueing conditions, dampening occurred more rapidly after four strides.

LDS and ACI are defined as slopes of the divergence curves measured at different intervals. Given the dampening, it was expected that ACI measured further from the initial separation would exhibit lower values. This was confirmed, as shown in the Fig. 3 boxplots: ACI [1–4] was higher and more variable than either ACI [4–7] or ACI [7–10]. Furthermore, the LDS, which was computed during the first step, was larger (Fig. 2).

As shown by the effect size plots in Fig. 4, ACIs decreased strongly when individuals followed auditory or visual cues. The effect was most pronounced for the AP signal, for

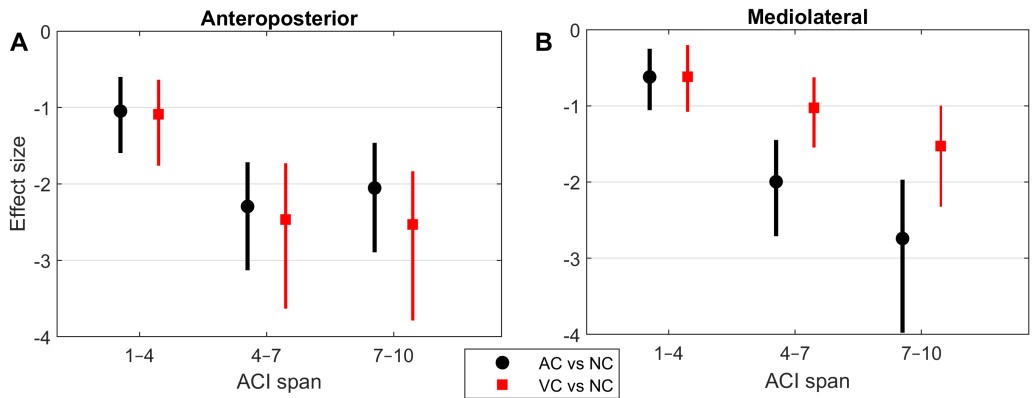

**Figure 4  Effect sizes of attractor complexity index (ACI).** Standardized effect size (Hedges' *g*) of the difference between cueing and no-cueing conditions for the anteroposterior (A) and the mediolateral signals (B). Vertical lines are 95% confidence intervals (Bonferroni corrected). AC, auditory cueing; VC, visual cueing; NC, no-cueing.

| LDS AP | | | | | | | | | | |
|---|---|---|---|---|---|---|---|---|---|---|
| r:-0.05 | **LDS ML** | | | | | | | | | |
| **r:-0.33** | r:0.16 | **ACI 1 AP** | | | | | | | | |
| r:-0.17 | r:0.03 | **r:0.71** | **ACI 2 AP** | | | | | | | |
| r:-0.08 | r:0.12 | **r:0.51** | **r:0.88** | **ACI 3 AP** | | | | | | |
| **r:-0.25** | r:-0.03 | **r:0.69** | **r:0.44** | **r:0.27** | **ACI 1 ML** | | | | | |
| **r:-0.20** | r:0.01 | **r:0.60** | **r:0.69** | **r:0.55** | **r:0.61** | **ACI 2 ML** | | | | |
| r:-0.15 | r:0.01 | **r:0.57** | **r:0.79** | **r:0.73** | **r:0.47** | **r:0.87** | **ACI 3 ML** | | | |
| r:-0.16 | r:0.10 | **r:0.56** | **r:0.86** | **r:0.82** | **r:0.42** | **r:0.74** | **r:0.84** | **Alpha ST** | | |
| **r:-0.20** | r:0.09 | **r:0.60** | **r:0.86** | **r:0.78** | **r:0.47** | **r:0.76** | **r:0.84** | **r:0.94** | **Alpha SL** | |
| **r:-0.31** | r:-0.02 | **r:0.46** | **r:0.44** | **r:0.32** | **r:0.46** | **r:0.39** | **r:0.40** | **r:0.27** | **r:0.51** | **Alpha SS** |

**Figure 5  Correlations and scatter plots across local dynamic stability (LDS), attractor complexity index (ACI), and scaling exponent (alpha) measures.** Pearson's correlation coefficients (*r*) are shown on the lower left. Bold values indicate significant results for the hypothesis test for *r* = 0. In the upper right, scatter plots with the linear fits are shown. AP, anteroposterior; ML, mediolateral; ST, stride time; SL, stride length; SS, stride speed.

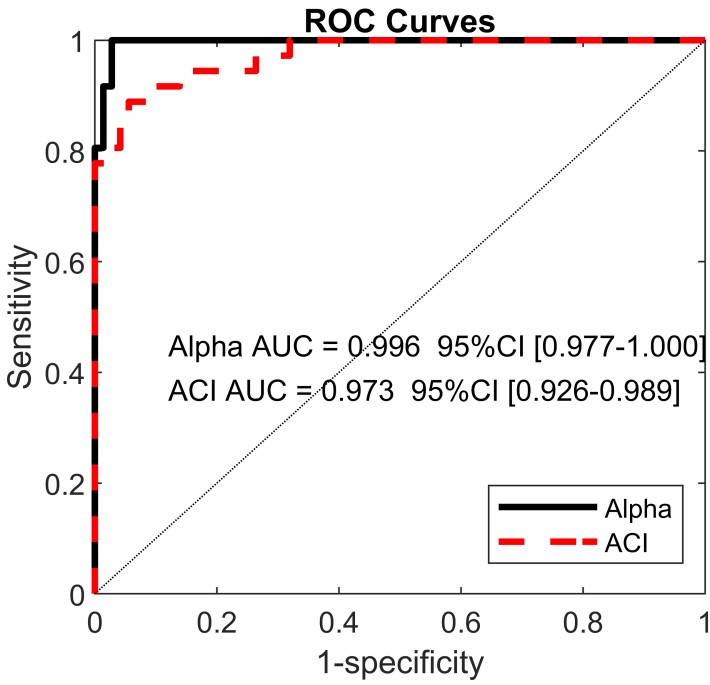

**Figure 6 Receiver operating characteristic (ROC) curves.** ROC curves for two multivariable logistic models predicting cueing/no-cueing conditions: (1) scaling exponent (alpha); and (2) attractor complexity index (ACI). Areas under the curves (AUCs) are written with their confidence intervals.

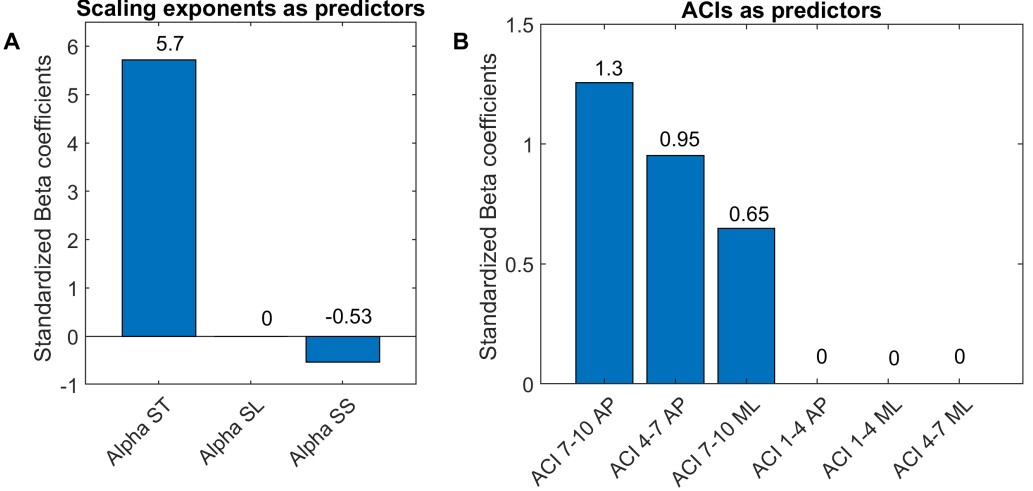

**Figure 7 Standardized coefficients of the multivariable logistic models.** Two multivariable logistic models were fitted using: (A) scaling exponents (Alphas); and (B) attractor complexity indexes (ACIs). A least absolute shrinkage and selection operator (LASSO) was used to regularize fitting. Bars show the value of the standardized beta coefficient of the logistic regressions for each predictor. AP, anteroposterior; ML, mediolateral; ST, stride time; SL, stride length; SS, stride speed.

which both auditory and visual cueing had comparable effects. In contrast, a relevant difference existed between auditory and visual cueing for the ML signals. Regarding LDS, the only significant effect was between the auditory and no-cueing conditions in the ML directions (standardized effect size: $-0.38$; 95% CI [$-0.69$–$-0.04$]).

Figure 5 shows the correlations among the LDS, ACI, and scaling exponents. Of particular note is the high correlation found between ACI [4–7] measured by the AP direction and the scaling exponents ($r = 0.86$ with $\alpha$-ST, and $r = 0.86$ with $\alpha$-SL). Other ACI spans exhibited weaker correlations. ML-LDS was not correlated with other variables, while AP-LDS was weakly and negatively correlated with scaling exponents ($r = -0.20$ with $\alpha$-ST, and $r = -0.30$ with $\alpha$-SS).

Using the ACIs and scaling exponents, multivariable logistic models differentiated very well between the cueing and no-cueing conditions. The AUCs were close to 1 ($\alpha$ AUC = 0.996, ACI AUC = 0.973; Fig. 6). ACI model's sensitivity was 94%, and specificity was 89%. In contrast, regarding LDS, the LASSO shrinkage procedure reduced the coefficients to zero, indicating a non-significant model.

As shown in Fig. 7, The LASSO algorithm selected the most significant predictors, and no important ones were set to 0. The LDS predictors are not shown, because all the coefficients were null. The strongest predictors were $\alpha$-ST and ACIs measured in the AP direction over long-term spans (4–10).

## DISCUSSION

The aim of this study was to further explore whether ACI could be used to assess gait complexity from continuous signals. The results strongly support the hypothesis that both DFA and ACI measure the same thing: their values were strongly correlated, they both differed strongly between the cueing and no-cueing conditions, and they both predicted cueing conditions with high degrees of sensitivity and specificity. The results also show that ACI should be measured in the AP direction and between four to seven strides downstream from the initial separation. In addition, LDS seemed insensitive to cueing, further supporting its use as a pure gait stability index.

A previous study assessed the effect of auditory cueing on stride-to-stride fluctuations in a treadmill experiment among 20 young adults (*Terrier & Dériaz, 2012*). Scaling exponents of SL and ST were strongly anti-persistent ($\alpha < 0.5$) under the auditory cueing condition. Based on the same data, another study investigated the effects of auditory cueing on LDS and ACI (*Terrier & Dériaz, 2013*). ACI (still referred to as $\lambda$-L at that time) was computed over a timescale between the 4th and 10th strides. The standardized effect size of the difference between the no-cueing and auditory cueing conditions was $-3.3$ for both the AP and ML signals. In addition, a substantial correlation between the scaling exponents and ACI was found (canonical correlation: $r = 0.83$). Another research group also found similar results in a study that combined a foot-switch and an accelerometer to evaluate overground walking (*Sejdić et al., 2012*); they found that both ACIs ($\lambda$-LT) and scaling exponents were substantially lower when the walk was paced with a metronome. The results of the present study confirm ACI's sensitivity to auditory cueing (effect size $<-2$;
Fig. 4). Overall, ACI seems sensitive to changes of long-range fluctuation patterns induced by auditory sensorimotor synchronization.

The influence of visual cueing on ACI had not been previously studied. The present results indicate that both visual and auditory cueing induced similar modifications to ACIs measured from the AP signal (Figs. 1 and 4). Previous research has also demonstrated that visual and auditory cueing have similar effects on scaling exponents (*Terrier, 2016*), which are incidentally computed from the discretization of the AP signal. In contrast, the present study found that when using ML measures, visual cueing had less of an effect than did auditory cueing (Fig. 4). It is worth noting that the visual cueing procedure consisted of participants aiming their feet toward rectangular visual targets (stepping stones). As a result, the task required voluntary leg control in both the AP and ML directions. Further analyses are needed to specifically explore gait lateral control under such circumstances, for instance by analyzing time series of step widths, which would be computed from the discretization of the ML signal (see *Terrier, 2012*).

LDS and ACI are rates of divergence (i.e., slopes) computed from an average logarithmic divergence curve (Fig. 1). Contrary to a real chaotic attractor, gait divergence curves do not exhibit a linear region, from which the slope should be computed according to the Rosenstein algorithm (*Rosenstein, Collins & De Luca, 1993*; *Terrier & Dériaz, 2013*). In fact, as illustrated in Fig. 1, the divergence rate diminishes continuously along the curve. The determination of range for computing ACI is therefore not straightforward. In their seminal researches, Dingwell et al. computed the slope between the 4th and 10th strides, but without a clear justification for this range (*Dingwell et al., 2000*; *Dingwell & Cusumano, 2000*). Subsequent studies followed identical spans. However, based on an examination of the divergence curves, it may be unnecessary to go that far from initial separation to estimate a meaningful long-term divergence, especially since this also increases computational cost. For instance, it was recently shown that an ACI (LDS-L) computed between the 2nd and 6th strides could discriminate between healthy individuals and patients suffering chronic pain of lower limbs (*Terrier et al., 2017*). In addition, the recent modeling study that introduced ACI observed that the ACI measured between the 2nd and 4th strides was more responsive to the stride-to-stride noise structure than the ACI measured between the 4th and the 10th strides, i.e., the originally proposed range (*Terrier & Reynard, 2018*). Here, the results show that ACI [4–7] was superior to the other ranges: it exhibited the highest correlation with the scaling exponents of ST and SL ($r = 0.86$ and $0.86$; Fig. 5), it had the highest contrast with the no-cueing condition (Fig. 4), and it was selected by the logistic model as the second highest predictor of cueing conditions (standardized coefficient = 0.95; Fig. 7). In short, it is very likely that it is not necessary to measure divergence after the 7th stride to assess ACI.

The results support the hypothesis that LDS and ACI measure different aspects of gait control. Notably, LDS was not able to predict cueing conditions (not significant logistic model), and most of the correlations between LDS and scaling exponents were weak (Fig. 5). The only significant correlations concerned the AP-LDS and they were negative, i.e., went in opposite direction compared to the ACI correlations. ML-LDS has been shown to be an index of gait instability (*Reynard et al., 2014*) and fall risk (*Bizovska et al., 2018*). This may be due to the importance of lateral stability for maintaining a steady and safe gait

(*Bauby & Kuo, 2000*; *Gafner et al., 2017*). The results of the present study support the use of ML-LDS for stability assessments given its total independence from complexity measures (Fig. 5). However, it is unclear whether results obtained from center-of-pressure trajectory are comparable to those obtained with other methods, such as trunk accelerometry; incidentally, a large-scale accelerometry study found that AP-LDS could predict future falls (*Van Schooten et al., 2015*). The assumption that ML-LDS is better suited for gait stability assessments thus requires further investigations. Overall, renaming long-term LDS as ACI is further legitimated given the distinct responsiveness of short- and long-term LDS to cueing.

The biggest strength of the present study is in its substantial number of strides measured in a large sample of healthy adults (36), particularly when compared to other recent studies in the field (*Sejdić et al., 2012*; *Bohnsack-McLagan, Cusumano & Dingwell, 2016*; *Roerdink et al., 2019*). Evaluating gait complexity requires the analysis a large number of consecutive strides (*Marmelat & Meidinger, 2019*). Similarly, reliability results show that many consecutive strides are required to accurately assess ACI (*Reynard & Terrier, 2014*). As far as I know, LDS and ACI have never been computed over a such large number of consecutive strides (500) so far. Consequently, this study's findings most likely offer good generalizability.

The study's primary limitation is that the analyses of the center-of-pressure trajectories are restricted to treadmill experiments with few potential applications. The center-of-pressure approach has the advantage of allowing an easy discretization to compare both discrete time series and continuous signals (*Roerdink et al., 2008*), but further investigations are required to explore ACI potential in real-life applications using inertial sensors such as accelerometers. Finally, it is important to underline that the use of ACI to assess the fluctuation structure of gait is purely based on empirical considerations and has no clear theoretical support for now. Further theoretical studies are required to investigate the relationship between scaling exponents and ACI.

## CONCLUSIONS

This study's findings support the hypothesis that ACI can provide information about the stride-to-stride fluctuation structure of an individual's gait based on continuous signals. Given that ACI fully harnesses continuous signals, it is not excluded that it requires fewer consecutive strides than DFA for an accurate measurement, but this requires further studies. Accordingly, information about gait complexity can be obtained while measuring a gait with inertial sensors, such as accelerometers (*Terrier et al., 2017*; *Terrier & Reynard, 2018*).

ACI could thus assess the degree of motor control applied by walkers on their gait (the "thigh control" hypothesis; see *Dingwell & Cusumano (2010)* and *Roerdink et al. (2019)*). A high ACI would indicate an automated gait, while a lower ACI would be a sign of greater voluntary attention dedicated to gait control. For example, it has been previously suggested that a low ACI in patients with lower limb pain is due to enhanced gait control to avoid putting too much weight on a painful leg (*Terrier et al., 2017*). Older studies that

inappropriately used ACI as a gait stability index should be reinterpreted with the "thigh control hypothesis" taken into account. For example, *Dingwell et al.(2000)* found that patients suffering from peripheral neuropathy had lower ACIs, which was interpreted as a higher gait stability obtained by lowering walking speed. An alternative explanation would be that diminished sensory feedback required more attention dedicated to gait control.

The use of LDS to characterize gait stability and assess fall risk has gained popularity over recent years (*Mochizuki & Aliberti, 2017*; *Bizovska et al., 2018*; *Mehdizadeh, 2018*). Computing ACI in addition to LDS can be made without further computation. Using ACI and LDS together could be fruitful, as information about gait automaticity and cautiousness would complement information about gait stability. It is hoped that the results of this study will help convince researchers to reinstate the use of ACI to further enrich their gait analysis studies.

### Funding
No external funding was received for this study.

### Competing Interests
The author declares that he has no competing interests.

### Author Contributions
- Philippe Terrier conceived and designed the experiments, performed the experiments, analyzed the data, contributed reagents/materials/analysis tools, prepared figures and/or tables, authored or reviewed drafts of the paper, approved the final draft.

### Data Availability
Terrier, Philipe (2019): Complexity of human walking: the attractor complexity index is sensitive to gait synchronization with visual and auditory cues. figshare. Dataset. https://doi.org/10.6084/m9.figshare.8166902.v2.

### Supplemental Information
Supplemental information for this article can be found online at http://dx.doi.org/10.7717/peerj.7417#supplemental-information.

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

# PeerJ

**Bauby CE, Kuo AD. 2000.** Active control of lateral balance in human walking. *Journal of Biomechanics* **33**:1433–1440 DOI 10.1016/S0021-9290(00)00101-9.

**Bizovska L, Svoboda Z, Janura M, Bisi MC, Vuillerme N. 2018.** Local dynamic stability during gait for predicting falls in elderly people: a one-year prospective study. *PLOS ONE* **13**:e0197091 DOI 10.1371/journal.pone.0197091.

**Bohnsack-McLagan NK, Cusumano JP, Dingwell JB. 2016.** Adaptability of stride-to-stride control of stepping movements in human walking. *Journal of Biomechanics* **49**:229–237 DOI 10.1016/j.jbiomech.2015.12.010.

**Bruijn SM, Bregman DJJ, Meijer OG, Beek PJ, Van Dieën JH. 2012.** Maximum Lyapunov exponents as predictors of global gait stability: a modelling approach. *Medical Engineering & Physics* **34**:428–436.

**Bruijn SM, Meijer OG, Beek PJ, Van Dieën JH. 2013.** Assessing the stability of human locomotion: a review of current measures. *Journal of the Royal Society, Interface* **10**:Article 20120999 DOI 10.1098/rsif.2012.0999.

**Chang MD, Sejdić E, Wright V, Chau T. 2010.** Measures of dynamic stability: detecting differences between walking overground and on a compliant surface. *Human Movement Science* **29**:977–986 DOI 10.1016/j.humov.2010.04.009.

**Choi J-S, Kang D-W, Seo J-W, Tack G-R. 2017.** Fractal fluctuations in spatiotemporal variables when walking on a self-paced treadmill. *Journal of Biomechanics* **65**:154–160 DOI 10.1016/j.jbiomech.2017.10.015.

**Delignières D, Torre K. 2009.** Fractal dynamics of human gait: a reassessment of the 1996 data of Hausdorff et al. *Journal of Applied Physiology* **106**:1272–1279 DOI 10.1152/japplphysiol.90757.2008.

**Dingwell JB. 2006.** Lyapunov exponents. In: *Wiley Encyclopedia of Biomedical Engineering.* Hoboken: John Wiley & Sons, Inc. DOI 10.1002/9780471740360.ebs0702.

**Dingwell JB, Cusumano JP. 2000.** Nonlinear time series analysis of normal and pathological human walking. *Chaos* **10**:848–863 DOI 10.1063/1.1324008.

**Dingwell JB, Cusumano JP. 2010.** Re-interpreting detrended fluctuation analyses of stride-to-stride variability in human walking. *Gait & Posture* **32**:348–353 DOI 10.1016/j.gaitpost.2010.06.004.

**Dingwell JB, Cusumano JP, Sternad D, Cavanagh PR. 2000.** Slower speeds in patients with diabetic neuropathy lead to improved local dynamic stability of continuous overground walking. *Journal of Biomechanics* **33**:1269–1277 DOI 10.1016/S0021-9290(00)00092-0.

**Fraser AM, Swinney HL. 1986.** Independent coordinates for strange attractors from mutual information. *Physical Review. A, Atomic, Molecular, and Optical Physics* **33**:1134–1140 DOI 10.1103/PhysRevA.33.1134.

**Gafner S, Bastiaenen C, Ferrari S, Gold G, Terrier P, Hilfiker R, Allet L. 2017.** Hip muscle and hand-grip strength to differentiate between older fallers and non-fallers: a cross-sectional validity study. *Clinical Interventions in Aging* **13**:1–8 DOI 10.2147/CIA.S146834.

**Goldberger AL, Amaral LAN, Hausdorff JM, Ivanov PC, Peng C-K, Stanley HE. 2002.** Fractal dynamics in physiology: alterations with disease and aging. *Proceedings of the*

*National Academy of Sciences of the United States of America* **99(Suppl 1)**:2466–2472 DOI 10.1073/pnas.012579499.

**González RC, López AM, Rodriguez-Uría J, Alvarez D, Alvarez JC. 2010.** Real-time gait event detection for normal subjects from lower trunk accelerations. *Gait & Posture* **31**:322–325 DOI 10.1016/j.gaitpost.2009.11.014.

**Hausdorff JM, Ladin Z, Wei JY. 1995.** Footswitch system for measurement of the temporal parameters of gait. *Journal of Biomechanics* **28**:347–351 DOI 10.1016/0021-9290(94)00074-E.

**Hausdorff JM, Peng CK, Ladin Z, Wei JY, Goldberger AL. 1995.** Is walking a random walk? Evidence for long-range correlations in stride interval of human gait. *Journal of Applied Physiology* **78**:349–358 DOI 10.1152/jappl.1995.78.1.349.

**Holt KG, Jeng SF, Ratcliffe R, Hamill J. 1995.** Energetic cost and stability during human walking at the preferred stride frequency. *Journal of Motor Behavior* **27**:164–178 DOI 10.1080/00222895.1995.9941708.

**Kennel MB, Brown R, Abarbanel HD. 1992.** Determining embedding dimension for phase-space reconstruction using a geometrical construction. *Physical Review. A, Atomic, Molecular, and Optical Physics* **45**:3403–3411 DOI 10.1103/PhysRevA.45.3403.

**Lopez AM, Alvarez D, Gonzalez RC, Alvarez JC. 2008.** Validity of four gait models to estimate walked distance from vertical COG acceleration. *Journal of Applied Biomechanics* **24**:360–367 DOI 10.1123/jab.24.4.360.

**Marmelat V, Meidinger RL. 2019.** Fractal analysis of gait in people with Parkinson's disease: three minutes is not enough. *Gait & Posture* **70**:229–234 DOI 10.1016/j.gaitpost.2019.02.023.

**McAndrew PM, Wilken JM, Dingwell JB. 2011.** Dynamic stability of human walking in visually and mechanically destabilizing environments. *Journal of Biomechanics* **44**:644–649 DOI 10.1016/j.jbiomech.2010.11.007.

**Mehdizadeh S. 2018.** The largest Lyapunov exponent of gait in young and elderly individuals: a systematic review. *Gait & Posture* **60**:241–250 DOI 10.1016/j.gaitpost.2017.12.016.

**Mochizuki L, Aliberti S. 2017.** Gait stability and aging. In: Barbieri FA, Vitório R, eds. *Locomotion and posture in older adults: the role of aging and movement disorders.* Cham: Springer International Publishing, 45–54 DOI 10.1007/978-3-319-48980-3_4.

**Peng CK, Buldyrev SV, Goldberger AL, Havlin S, Mantegna RN, Simons M, Stanley HE. 1995.** Statistical properties of DNA sequences. *Physica A* **221**:180–192 DOI 10.1016/0378-4371(95)00247-5.

**Pereira APS, Marinho V, Gupta D, Magalhães F, Ayres C, Teixeira S. 2019.** Music therapy and dance as gait rehabilitation in patients with parkinson disease: a review of evidence. *Journal of Geriatric Psychiatry and Neurology* **32**:49–56 DOI 10.1177/0891988718819858.

**Reynard F, Terrier P. 2014.** Local dynamic stability of treadmill walking: intrasession and week-to-week repeatability. *Journal of Biomechanics* **47**:74–80 DOI 10.1016/j.jbiomech.2013.10.011.

**Reynard F, Vuadens P, Deriaz O, Terrier P. 2014.** Could local dynamic stability serve as an early predictor of falls in patients with moderate neurological gait disorders? A reliability and comparison study in healthy individuals and in patients with paresis of the lower extremities. *PLOS ONE* **9**:e100550 DOI 10.1371/journal.pone.0100550.

**Riva F, Toebes MJP, Pijnappels M, Stagni R, Van Dieën JH. 2013.** Estimating fall risk with inertial sensors using gait stability measures that do not require step detection. *Gait & Posture* **38**:170–174 DOI 10.1016/j.gaitpost.2013.05.002.

**Roerdink M, Coolen BH, Clairbois BHE, Lamoth CJC, Beek PJ. 2008.** Online gait event detection using a large force platform embedded in a treadmill. *Journal of Biomechanics* **41**:2628–2632 DOI 10.1016/j.jbiomech.2008.06.023.

**Roerdink M, Daffertshofer A, Marmelat V, Beek PJ. 2015.** How to sync to the beat of a persistent fractal metronome without falling off the treadmill? *PLOS ONE* **10**:e0134148 DOI 10.1371/journal.pone.0134148.

**Roerdink M, De Jonge CP, Smid LM, Daffertshofer A. 2019.** Tightening up the control of treadmill walking: effects of maneuverability range and acoustic pacing on stride-to-stride fluctuations. *Frontiers in Physiology* **10**:Article 257 DOI 10.3389/fphys.2019.00257.

**Rosenstein MT, Collins JJ, De Luca CJ. 1993.** A practical method for calculating largest Lyapunov exponents from small data sets. *Physica D: Nonlinear Phenomena* **65**:117–134 DOI 10.1016/0167-2789(93)90009-P.

**Sejdić E, Fu Y, Pak A, Fairley JA, Chau T. 2012.** The effects of rhythmic sensory cues on the temporal dynamics of human gait. *PLOS ONE* **7**:e43104 DOI 10.1371/journal.pone.0043104.

**Su JL-S, Dingwell JB. 2007.** Dynamic stability of passive dynamic walking on an irregular surface. *Journal of Biomechanical Engineering* **129**:802–810 DOI 10.1115/1.2800760.

**Takens F. 1981.** Detecting strange attractors in turbulence. In: Rand D, Young L-S, eds. *Dynamical systems and turbulence, Warwick 1980. Lecture notes in mathematics*, Berlin Heidelberg: Springer, 366–381.

**Terrier P. 2012.** Step-to-step variability in treadmill walking: influence of rhythmic auditory cueing. *PLOS ONE* **7**:e47171 DOI 10.1371/journal.pone.0047171.

**Terrier P. 2016.** Fractal fluctuations in human walking: comparison between auditory and visually guided stepping. *Annals of Biomedical Engineering* **44**:2785–2793 DOI 10.1007/s10439-016-1573-y.

**Terrier P, Carré JL, Connaissa M, Léger B, Luthi F. 2017.** Monitoring of gait quality in patients with chronic pain of lower limbs. *IEEE Transactions on Neural Systems and Rehabilitation Engineering* **25**:1843–1852 DOI 10.1109/TNSRE.2017.2688485.

**Terrier P, Dériaz O. 2011.** Kinematic variability, fractal dynamics and local dynamic stability of treadmill walking. *Journal of NeuroEngineering and Rehabilitation* **8**:Article 12 DOI 10.1186/1743-0003-8-12.

**Terrier P, Dériaz O. 2012.** Persistent and anti-persistent pattern in stride-to-stride variability of treadmill walking: influence of rhythmic auditory cueing. *Human Movement Science* **31**:1585–1597 DOI 10.1016/j.humov.2012.05.004.
**Terrier P, Dériaz O. 2013.** Non-linear dynamics of human locomotion: effects of rhythmic auditory cueing on local dynamic stability. *Frontiers in Physiology* **4**:230 DOI 10.3389/fphys.2013.00230.

**Terrier P, Reynard F. 2015.** Effect of age on the variability and stability of gait: a cross-sectional treadmill study in healthy individuals between 20 and 69 years of age. *Gait & Posture* **41**:170–174 DOI 10.1016/j.gaitpost.2014.09.024.

**Terrier P, Reynard F. 2018.** Maximum Lyapunov exponent revisited: long-term attractor divergence of gait dynamics is highly sensitive to the noise structure of stride intervals. *Gait & Posture* **66**:236–241 DOI 10.1016/j.gaitpost.2018.08.010.

**Terrier P, Schutz Y. 2005.** How useful is satellite positioning system (GPS) to track gait parameters? A review. *Journal of Neuroengineering and Rehabilitation* **2**:Article 28 DOI 10.1186/1743-0003-2-28.

**Terrier P, Turner V, Schutz Y. 2005.** GPS analysis of human locomotion: further evidence for long-range correlations in stride-to-stride fluctuations of gait parameters. *Human Movement Science* **24**:97–115 DOI 10.1016/j.humov.2005.03.002.

**Tibshirani R. 1996.** Regression shrinkage and selection via the Lasso. *Journal of the Royal Statistical Society: Series B (Methodological)* **58**:267–288 DOI 10.1111/j.2517-6161.1996.tb02080.x.

**Van Schooten KS, Pijnappels M, Rispens SM, Elders PJM, Lips P, Van Dieën JH. 2015.** Ambulatory fall-risk assessment: amount and quality of daily-life gait predict falls in older adults. *The Journals of Gerontology. Series A, Biological Sciences and Medical Sciences* **70**:608–615 DOI 10.1093/gerona/glu225.

**Van Schooten KS, Sloot LH, Bruijn SM, Kingma H, Meijer OG, Pijnappels M, Van Dieën JH. 2011.** Sensitivity of trunk variability and stability measures to balance impairments induced by galvanic vestibular stimulation during gait. *Gait & Posture* **33**:656–660 DOI 10.1016/j.gaitpost.2011.02.017.

**West BJ. 2013.** *Fractal physiology and chaos in medicine.* New Jersey: World Scientific.

**Yoo GE, Kim SJ. 2016.** Rhythmic auditory cueing in motor rehabilitation for stroke patients: systematic review and meta-analysis. *Journal of Music Therapy* **53**:149–177 DOI 10.1093/jmt/thw003.