# Peer review of "Complexity of human walking: the attractor complexity index is sensitive to gait synchronization with visual and auditory cues"

_PeerJ, doi:10.7717/peerj.7417_

## Round 0.1 · original submission · Major Revisions

The reviewers generally commented positively and provided feedback to further strengthen the manuscript. In particular, the authors should clarify the benefits of the proposed measure over existing measures and address the potential bias of using different number of strides. Also, I encourage the authors to share the raw data.

·

Basic reporting

Review: PeerJ#37165
Title: Complexity of human walking: the attractor complexity index is sensitive to gait synchronization with visual and auditory cues

This manuscript describes a study undertaken to assess a novel measure of complexity of gait, which is compared to DFA. To this aim, the attractor complexity index is introduced, which is basically the rate of divergence (as used for local dynamic stability calculations, also sometimes referred to as Lyapunove exponents) after different tracking times. It is shown that a state space constructed from AP signals yields ACI which are highly correlated to DFA results, and also show similar changes with cueing. All in all, this is a well written and interesting manuscript. I have only two major comments, which may require re-analysis of the data.

Experimental design

Major comment:

1) If I understand correctly, for DFA, all steps measures were included in the calculation (at leats, there is no mention of a specific selection of strides), whereas for ACI and LDS calculations, only 300 strides were used. While this is OK, it could well be that if DFA and ACI (and LDS) were calculated over the same strides (which is not the case now, as DFA is calculated over more strides), agreement between the two measures would be even higher. It may thus be good to redo the calculations, such that all measures are calculated from exactly the same data.

Validity of the findings

See 2.

Additional comments

Major comment:

2) While data are shared, I do not consider these the “raw” data, as they are simply the calculated parameters. There can of course be debate about whether that constitutes “raw” data or not, but I feel it my duty to at least mention this here.

Minor comments:
1) Please avoid the use of abbreviations for the conditions. This makes reading unnecessary hard for readers. PeerJ does not have strict word limits, so no need to make your readers suffer through the abbreviations.
2) In the abstract, an Area under the curve of 1 is mentioned, but this cannot be found in the results?
3) Figure 5 is rather unreadable; maybe it can be made somewhat bigger?

Reviewer 2 ·

Basic reporting

Abstract:
- It would be good if the author provides an explanation on what would be the benefits of using the new measure (ACI) that uses continuous time series over DFA in terms of extra information it can provide.
- I would suggest using AUC for the abbreviation of the area under the ROC curve as it is more common.
- If ACI and DFA perform equally, then what is the point of using ACI. Please refer to my first comment.
Introduction:
- I commend the authors for the comprehensive background on DFA.
- Line 94-101- how the interpretation of alpha in its classical form is changed according to the study of Dingwell and Cusumano, Gait Posture. 2010 Jul;32(3):348-53. doi: 10.1016/j.gaitpost.2010.06.004. I suggest discussing this paper in this paragraph as well.
- I still don’t get what extra benefit ACI provides over DFA alpha. I think more rationale is needed to convince the reader.
- In addition, more theoretical background is needed on the relevance of ACI to DFA from a theoretical point of view as ACI is simply the long-term LDS.

Experimental design

- Line 183- doesn’t filtering affect the calculation of ACI and DFA? Any reference?
- Line 189- 192- please cite appropriately for delay embedding and average mutual information method. Also the Rosenstein’s study
- Line 219-p=0.5 or 0.05?
- According to Figure 1, it doesn’t seem that AC1-4 (slope over strides 1 to 4) is a good linear fit of the region? What is the rationale for this measure?

Validity of the findings

- The findings are valid and support the hypothesis. But my concern again is the theoretical connection between ACI and DFA.
- In a similar manner, is there any ACI value equivalent to alpha=0.5?
- As also mentioned by the author in the Introduction, previous studies didn’t find different long-term LDS between stability condition. What is the difference between the method of calculating ACI in this study and previous ones that makes it unique to be used as a fractal measure? Considering that it can differentiate between conditions in this study. This becomes more important considering that short-term LDS in AP could efficiently differentiate the conditions in this study with AUC=0.82 implying that short- and long-term LDS are measuring the same thing (which indeed they are)

Reviewer 3 ·

Basic reporting

The manuscript is clear overall. The figures looks clear, although I suggest to shift x and y axis in Figure 2 and 3. In Figure 2 I suggest to use the same scale for both ML and AP LDS, as it is it seems that in No cueing they both present the same range but AP LDS is much wider. While a 'Raw Data' file was provided, it appears it contains only the main outcomes for the analysis, not the actual raw data necessary if one would like to replicate the author's results.

Experimental design

A few more details are necessary in the Methods section: notably, while I agree that a full description of DFA is not necessary, it is crucial to report the parameters selected (e.g., n_min and n_max, evenly spaced algorithm or not?). While I understand that the DFA results can be found in previous publication, I think enough information should be provided in the present article to allow replication.

Validity of the findings

The results may shed new lights on previously published work investigating long-term LDS (or ACI). Overall the conclusions are supported by the results, although access to the raw time series would benefit potential for replication.

---

## Round 0.2 · Minor Revisions

The reviewers have some additional minor comments that need to be addressed before the manuscript can be accepted for publication.

Reviewer 2 ·

Basic reporting

the author covered almost all comment. The only remaining comment is:

in response to my previous comment "In addition, more theoretical background is needed on the relevance of ACI to DFA from a theoretical point of view as ACI is simply the long-term LDS." he replied: "The reason of the sensitivity of ACI to long-term gait complexity is still an enigma for me. I cannot provide more information than that I gave in the introduction (L 162-168). I hope that specialists in nonlinear gait analysis can further investigate this point."

he also continues to another comment "...the sensitivity of ACI to noise structure of stride intervals is purely based on observations and has no theoretical justification for now."

These responses confirm that the results of this study are based on observations and future studies are needed to investigate the theoretical relationship between DFA and ACI. this should be added and underlined in the Abstract and Conclusion.

Experimental design

The author covered all comment

Validity of the findings

the author covered all comment

Reviewer 3 ·

Basic reporting

The authors has made substantial revisions to address each reviewer's comments. The figures are clearer, but most important the rationale as to why it is important to compare LDS (ACI) to DFA measures is better introduced and discussed.
I also appreciate that the author does not try to answer some questions with some convoluted explanations, but simply state 'I don't know'.

Experimental design

The revisions in the Methods (comparing two techniques on the 'same' signal) certainly enhance the potential impact. I am personally curious to see if the results holds if ACI is computed from shorter time series; in particular, if the relationship between ACI 2 (and ACI 3) with Alpha-DFA is still present. One of the limitations of DFA (and similar techniques) is that it requires lots of stride to obtain reliable estimates, which can limit its use for pathological populations. But if DFA and ACI are highly correlated even when ACI is computed on shorter time series, this may open the door to analyzing gait dynamics in under-studied populations (granted, a correlation between DFA and ACI does not necessarily mean they provide the same information!).
Note that I do NOT recommend this analysis in the present paper (although I think it may add another layer of information!). I just think it is something the author might consider in the future.

Validity of the findings

The addition of the actual raw data is very much appreciated.

Additional comments

Very good paper in general, I do not have more comments at this time. I would like to apologize to the editor and the author for being a bit late in my review.

---

## Round 0.3 · accepted · Accept

The author addressed the outstanding comments.